# Characterization of the NAC Transcription Factor in Passion Fruit (*Passiflora edulis*) and Functional Identification of *PeNAC-19* in Cold Stress

**DOI:** 10.3390/plants12061393

**Published:** 2023-03-21

**Authors:** Yi Xu, Pengfei Li, Funing Ma, Dongmei Huang, Wenting Xing, Bin Wu, Peiguang Sun, Binqiang Xu, Shun Song

**Affiliations:** 1State Key Laboratory of Biological Breeding for Tropical Crops, Tropical Crops Genetic Resources Institute, Chinese Academy of Tropical Agricultural Sciences, Germplasm Repository of Passiflora, Hainan Province, Hainan 571101, China; 2Sanya Research Institute, Chinese Academy of Tropical Agricultural Sciences, Sanya 571101, China; 3Hainan Yazhou Bay Seed Laboratory, Sanya 571101, China; 4College of Tropical Crops, Yunnan Agricultural University, Kunming 650201, China

**Keywords:** NAC transcription factor, passion fruit, abiotic stress, fruit maturity stage, gene expression

## Abstract

The NAC (NAM, ATAF and CUC) gene family plays an important role in plant development and abiotic stress response. However, up to now, the identification and research of the NAC (*PeNAC*) family members of passion fruit are still lacking. In this study, 25 *PeNACs* were identified from the passion fruit genome, and their functions under abiotic stress and at different fruit-ripening stages were analyzed. Furthermore, we analyzed the transcriptome sequencing results of *PeNACs* under four various abiotic stresses (drought, salt, cold and high temperature) and three different fruit-ripening stages, and verified the expression results of some genes by qRT-PCR. Additionally, tissue-specific analysis showed that most *PeNACs* were mainly expressed in flowers. In particular, *PeNAC-19* was induced by four various abiotic stresses. At present, low temperatures have seriously endangered the development of passion fruit cultivation. Therefore, *PeNAC-19* was transformed into tobacco, yeast and Arabidopsis to study their function of resisting low temperature. The results show that *PeNAC-19* responded to cold stress significantly in tobacco and Arabidopsis, and could improve the low temperature tolerance of yeast. This study not only improved the understanding of the *PeNAC* gene family characteristics and evolution, but also provided new insights into the regulation of the *PeNAC* gene at different stages of fruit maturation and abiotic stresses.

## 1. Introduction

Transcription factors (TFs) play an important role in regulating cell signaling, cell morphogenesis and plant resistance to external environmental stresses [1,2]. TFs regulate gene expression by binding to specific promoter cis-acting elements to activate or repress the transcriptional level of target genes [3,4]. In plants, the NAC (NAM, ATAF1/2 and CUC2) transcription factor family is named after three proteins: petunia apical meristem (NAM), *Arabidopsis thaliana* ATAF1/2 and cup cotyledon (CUC) [5,6], which is one of the largest and most plant-specific TF families. Typical NAC proteins include a highly conserved N-terminal region (NAC domain), while the C-terminal region contains a relatively distinct transcriptional activation/repression region (TAR or TRR) [5,7,8], which is a highly diverse transcriptional regulatory region [9], may be involved in protein–protein interactions and contribute to its regulatory specificity [10]. The N-terminus of the NAC protein contains a conserved domain of 150–160 amino acids involved in DNA binding, dimerization and localization [11], which is further divided into five subdomains (A-E), of which the A, C and D subdomains are highly conserved [12].

Since the first report of the NAC protein in 1996 [5], NAC protein families have been identified in several plant species, such as *Arabidopsis thaliana* [13], *Oryza sativa* [14], *Musa acuminata* [15], *Medicago sativa* [16], *Dimocarpus longan* [17], *Actinidia chinensis* [18], *Populus trichocarpa* [19], *Vitis vinifera* [20] and *Pyrus pyrifolia* [21]. Due to their powerful functions in plants, the NAC family has been extensively studied in recent years. The related reports indicate that NAC family members play an essential role in response to plant abiotic stress.

The overexpression of *OsNAC10* in rice can improve the drought tolerance and grain yield of plants [22]. The overexpression of *OsNAC6/SNAC2* can improve the drought tolerance, salt tolerance and cold tolerance of rice seedlings [23,24]. The transcription factor *MbNAC25* of Siberian crab apple (*Malus baccata* Borkh) can improve cold tolerance in transgenic Arabidopsis [25]. Both *SNAC2* of rice and *PbeNAC1* of pear (*Pyrus betulifolia* Bunge) confer cold tolerance in rice and pears [26]. In tomatoes, *SlNAM1* is induced by chilling stress and can improve the chilling resistance in transgenic plants [27].

In addition, there are also related studies showing that the NAC transcription factor is related to fruit development and ripening. Fruit development and ripening are complex processes regulated by various factors such as gene regulation, hormones, light and temperature The processes in the development and ripening of fruit are regulated by many factors such as light, temperature, hormone and gene regulation. In particular, they are regulated by a variety of transcription factors that affect the expression levels of downstream target genes [28]. Some studies have shown that NAC transcription factors can regulate the expression of the genes involved in hormone biosynthesis and signal transduction during fruit development and maturation [29]. In spruce, the overexpression of *PaNAC03* affects plant embryonic development [30]. Furthermore, in *Vitis vinifera*, the development of seeds and fruits is affected by the interaction between *VvNAC26* and *VvMADS9* [31]. In tomatoes, *SlNAC1* plays an important role in the softening process [32]. In strawberries (*Fragaria chiloensis*), the FcNAC1 protein is involved in pectin metabolism to soften fruit [33].

Passion fruit (*Passiflora edulis* Sim) is a perennial evergreen vine of the *Passiflora* genus of Passifloraceae, a tropical and rare fruit tree, which contains more than 100 kinds of aroma in its fruit pulp. It is native to central and northern South America, and is widely distributed throughout America, Australia and Africa [34]. According to the reports, *Passiflora* has about 520 varieties, most of which are used for ornamental purposes, and only a small number of 60 species can be eaten [35]. At present, the main countries for passion fruit are Brazil, Colombia, Ecuador, Australia, Vietnam, China, etc. Due to its unique flavor and short growth period (4–6 months), the planting area under cultivation gradually increased. Abiotic stresses seriously affect the normal development of the passion fruit industry. Therefore, it is important to excavate the function genes of stress resistance in passion fruit and analyze their mechanisms of action [36]. In this study, using the high-quality genomic data of passion fruit [37], 25 members of the passion fruit NAC (*PeNAC*) family were identified. In contrast to another previously published result [38], they used another genome [39]. The result of genome assemblies vary widely in the HR genome [37] and the MER genome [39]: contig N50 (3.1 Mb and 70 kb), complete BUSCOs (91.56% and 88.1%) and scaffold N50 (148,138.5 Mb and 126.4 Mb). More importantly, we also identified the expression patterns of this gene in different fruit-ripening stages and abiotic stresses, and validated them by qPCR. Additionally, one of the *P**eNAC* genes exhibited resistance to cold stress. These results provide useful information for the genetic improvement of fruit quality and the improvement of the abiotic stress resistance of passion fruit, and lay a good foundation for the study of the regulation mechanism of fruit quality.

## 2. Results

### 2.1. Identification of the Passion Fruit NAC Family

In this research, 25 *PeNACs* have been identified. In addition, we analyzed the characteristics of the *PeNACs* (Table 1). The length of the *PeNAC* CDS sequence ranged from 174 bp (*PeNAC-21*) to 6126 bp (*PeNAC-12*). The identified *PeNACs* encoded proteins ranging from 57 amino acids of *PeNAC-21* to 2041 amino acids of *PeNAC-12*. The MW ranged from 6.74Da (*PeNAC-21*) to 229.5Da (*PeNAC-12*). The isoelectric point ranged from 3.84 (*PeNAC-9*) to 10.44 (*PeNAC-7*). Subcellular localization predicted that all genes were located in the nucleus.

### 2.2. Phylogenetic Analysis of PeNACs Protein

To study the classification and evolutionary relationships of NAC proteins in passion fruit, a phylogenetic tree was constructed by the protein sequences of 25 *PeNACs* (passion fruit), 29 *AtNACs* (Arabidopsis) and 23 *OsNACs* (rice) (Figure 1). According to the kinship of the members, *PeNACs* were divided into three subfamilies: group 1 (*PeNAC-2/4/5/8/9/14/16/18/20/22*), group 2 (*PeNAC-1/10/12/17/24/25*) and group 3 (*PeNAC-3/6/7/11/13/15/19/21/23*). 

The homologous genes of NAC in passion fruit and Arabidopsis can be inferred due to the fact that passion fruit and Arabidopsis are both dicotyledones. For group 1, *PeNAC-4* was the best orthology matches of At1G56010.2. *PeNAC-8* was the most homogeneous genes of At3G01600.1. *PeNAC-9* were phylogenetically closest to At3G10490.3 and At3G10490.4. *PeNAC-14* exhibited the closest relationship with At1G25580.1. For group 2, *PeNAC-1* was the best orthology match of At3G46565.1. *PeNAC-12* was the most homogeneous gene of At1G32770.1 and At2G46770.1. For group 3, *PeNAC-21* was the best orthology match of At1G12260.2. *PeNAC-11* was phylogenetically closest to At3G18400.1. *PeNAC-15* exhibited the closest relationship with At5G41090.1 and At3G56520.1. 

### 2.3. Expression Pattern of NACs in Passion Fruit

The result of the RNA-seq data showed that the NACs have different response degrees to various abiotic stresses (Figure 2). Nineteen genes were induced by drought stress. The expression of *PeNAC-2/5/10/14/15/17/18/25* reached the highest levels when the soil water content was 10%. The expression level of some genes is induced to increase with the degree of salt stress, such as *PeNAC-2/3/10/11/16/23/25.* The transcript levels of *PeNAC-1/6/7/8/13* did not change much with the increase in salt stress. Under cold stress, most genes were upregulated. Among them, the expression of *PeNAC-1/3/4/6/7/13/20* displayed significant changes. Under high temperature stress, the expression of the *PeNAC-1/4/5/8/9/11/16/17/18/19/24* increased with the degree of stress. The expression of some genes (*PeNAC-6/7/10/13/14/15/20*) displayed no visible change. The results show that most of the *PeNACs* can respond to various abiotic stresses (Figure 3). 

We performed transcript sequencing during three periods: the seventh day before fruit harvest (Time1), the harvest period (Time2) and the seventh day after fruit harvest (Time3) in passion fruit [37], wherein we analyzed the expression levels of the *PeNACs* (Figure 4). The results show that most of the genes demonstrated the highest expression levels at Time1, and their expression decreased as the fruit matured, with the lowest expression in the third period, such as *PeNAC3/8/17/20/23/24/25*. Among them, *PeNAC22* is most expressed in the third period of fruit maturity. Some genes were verified by qRT-PCR and the results were consistent with the transcript sequencing (Figure 5). This indicated that this gene may be associated with fruit maturity.

The expression of some *PeNACs* in different tissues of passion fruit has been analyzed (Figure 6). Among them, four were mainly expressed in the flower (*PeNAC-6*, *PeNAC-17*, *PeNAC-19*, *PeNAC-23*). *PeNAC-1* was mainly expressed in the fruit. Additionally, *PeNAC-4/13/20* were mainly expressed in the root. The results show that the gene was expressed in different parts.

### 2.4. Cold Stress Analysis in Transgenic Tobacco

We further transiently transformed tobacco with the promoter of *PeNAC-19* (Figure 7). The *PeNAC-19p*-transformed tobacco and control were subjected to low-temperature stress treatment at 4 °C. The results show that the gene was highly induced during 2 h and 24 h treatments.

### 2.5. Functional Complementation Validation of PeNAC-19

In Figure 8, the pYES2-*PeNAC-19* and the pYES2 empty vector (control) were transformed into INVSC1 (*Saccharomyces cerevisiae*) for the cold-stress experiment (−20 °C for 0 h, 24 h, 48 h and 72 h). The results indicate that the growth of both the empty vector and the transgenic yeast is more and more restricted with increasing treatment time. When treated at −20 °C for 72 h, the *PeNAC-19*-transformed yeast grew better than the control. This suggests that *PeNAC-19* plays a certain role under the cold stress.

### 2.6. Response of Transgenic Arabidopsis to Low-Temperature Stress

As shown in Figure 9A, under normal growth conditions, GUS staining was mainly concentrated in stems, and under low-temperature stress, GUS staining was enhanced and mainly concentrated in the leaves and roots of the seedling. The GUS enzyme activity test was carried out, and we found that GUS activities are 3.4-fold higher than the control. The results show that *PeNAC-19* was induced by low-temperature stress.

## 3. Materials and Methods

### 3.1. Identification of NAC Genes in Passion Fruit

The passion fruit genome data were downloaded from the NGDC database (https://ngdc.cncb.ac.cn/gwh/Genome/557/show, accessed on 2 April 2021). Additionally, the PeNACs were initially screened and identified using hmmsearch and local blast (Output E value: 1 × 10^−5^). In addition, analysis was performed using the NAC with the highest comparison value to further identify possible NACs in the passion fruit database. The protein sequences identified by the two methods described above were integrated and resolved to remove redundancy. NAC protein sequences in Arabidopsis and rice were downloaded from databases (http://plants.ensembl.org/Arabidopsis_thaliana/Info/Index, accessed on 2 June 2022, http://plants.ensembl.org/Oryza_sativa/Info/Index, accessed on 3 June 2022). The phylogenetic trees were constructed using MEGA 7.0 software (https://www.megasoftware.net/, accessed on 12 June 2022) [40]. The members of *PeNACs* were finally obtained through the screening of the above methods.

### 3.2. Gene Identification

The physical and chemical properties of protein were calculated using the ProtParam website (http://www.expasy.org/tools/protparam.html, accessed on 15 June 2022) (https://meme-suite.org/meme/tools/meme, accessed on 20 June 2022), NetPhos 3.1 Server (http://www.cbs.dtu.dk/services/NetPhos/, accessed on 22 June 2022) and WoLF PSORT 0.2 software (https://www.genscript.com/wolf-psort.html, accessed on 29 June 2022). 

### 3.3. Plant Materials and Growth Conditions

For the abiotic stress treatment, the two-months-old healthy passion fruit seedlings (*Passiflora edulis*) were chosen. They were grown in soil under a growth chamber (30 °C; 200 µ mol·m^−2^·s^−1^ light intensity; 12 h light/12 h dark cycle; 70% relative humidity) to a height of about 1 m and with 8–10 functional leaves, which were subjected to various abiotic stress treatments [36]. For the sampling of fruit during ripening, the fruits were chosen in the 7th day before fruit harvest (Time1), the harvest period (Time2), and the 7th day after fruit harvest (Time3) in passion fruit [37]. Each treatment was repeated three times with 10 plants/fruits at a time, and the material was RNA extracted and RNAseq.

### 3.4. Heat Map Construction

Transcriptome data used for heat map construction are shown in Appendix A. The analysis was conducted using TBtools software (https://bio.tools/tbtools, accessed on 10 July 2022) [40].

### 3.5. Cloning and Vector Construction of PeNAC-19 and the Promoter

The full-length cDNA of *PeNAC-19* and a 2000 bp DNA fragment before the start codon of the *PeNAC-19* was amplified from the passion fruit (*Passiflora edulis*). Healthy passion fruit seedlings were used to construct a single-stranded cDNA template. 

To examine whether *PeNAC-19* responds to low-temperature stress, the expression vectors have been constructed. The PCR products of the *PeNAC-19* ORF were inserted into the pCAMBIA1304 (Cambia, Australia) expression vector, which is called pCAMBIA1304-*PeNAC-19.* The PCR products of the *PeNAC-19* promoter were cloned into the pCAMBIA1304 vector called pCAMBIA1304-*PeNAC-19p.* The vectors were transferred into the *Agrobacterium* strain EHA105.

### 3.6. Cold Stress Treatment in Wild-Type and Transgenic Plants

The two-month-old tobacco leaves were used for transient expression experiments. The *Agrobacterium* transformed with pCAMBIA1304-*PeNAC-19*p was shaken at 28 °C to OD 600 = 0.8–1.0, and the bacterial solution was infected with an equal volume of MES buffer (10 mM MES; 10 mM MgCl_2_; 200 mM MAS) and then kept in the dark at room temperature for 2–3 h. After that, the solution had been injected into the back of the tobacco which was left to culture for 2–3 d. The plants were treated at 4 °C and 25 °C for 0, 2, 4, 12 and 24 h. The leaf discs with a diameter of 0.5 cm were cut out for further staining testing.

The *Agrobacterium* transformed with pCAMBIA1304-*PeNAC-19* and pCAMBIA1304-*PeNAC19*p were shaken at 28 °C in YEB medium, and then added to 1/2 MS solution until OD 600 = 0.8–1.0. The transformation of *Arabidopsis thaliana* was conducted using the inflorescence dip method. The seeds of *Arabidopsis thaliana* were disinfected with 75% ethanol and spreading on the medium (1/2MS, 25 μg/mL hygromycin B). After culturing in the dark at 4 °C for 3 days, the seeds were transferred to a culture at 23 °C for growth. After ten days, the normal growth seedlings were the T1 generation. Additionally, the T2 generation grown in the selection medium were for further experimental analysis. The 14-day-old transgenic Arabidopsis with about 6–8 leaves was treated in an incubator at 4 °C and 23 °C for 36 h.

### 3.7. GUS Activity Detection

The fresh transgenic tobacco and Arabidopsis samples under the normal and cold conditions were placed in X-Gluc solution (GBT, St. Louis, MO, USA) [41] for histochemical analysis. Gus enzyme activity was determined by 4-methyl umbelliferate glucuronide fluorescence method [42].

### 3.8. Functional Complementation of PeNAC-19 in Yeast

The pYES2–*PeNAC-19* vector was constructed using the full-length cDNA of *PeNAC-19.* Then, the pYES2–*PeNAC-19* and pYES2 vector (control) were transfected into the INVSc1 Strain (*Saccharomyces cerevisiae*). To perform the yeast complementation assays, the yeast liquid was cultured in SD-Ura liquid medium at 30 °C until OD600 = 1.0 and treated under the cold stress [43]. The experiment was repeated three times.

### 3.9. RNA Extraction, Transcriptome Sequencing and qRT-PCR

Total RNA was extracted from frozen samples with plant RNA isolation kit (Fuji, China, Chengdu), and three biological replicates were performed. The cDNA was used for transcriptome sequencing analysis and quantitative real-time polymerase chain reaction (qRT-PCR). The SYBR^®^ Premix Ex Taq™ kit (TaKaRa, Tokyo, Japan) was used to detect in qRT-PCR. Relative expression levels were calculated using the 2^−∆∆CT^ method and normalized to the *PeNACs*.

## 4. Discussion

Passion fruit (*Passiflora edulis* Sim), a perennial vine, is rich in nutrients and contains a variety of amino acids and vitamins. Due to its short growth cycle and good economic value, it is very popular [36,37]. However, only a few studies have investigated its function [44]. Previous studies have shown that transcription factors can regulate plant growth and development and enhance plant responses to abiotic stresses [45]. The NAC family, which is one of the largest transcription factor families, plays an important role in plant growth, development and abiotic stress responses [46].

In this study, we have reported 25 *PeNACs* in passion fruit, and performed the expression patterns analysis under different abiotic stresses and during different fruit-ripening stages. At the same time, a low-temperature stress-inducible gene, *PeNAC-19*, was screened for transient expression in tobacco, transgenic Arabidopsis and in vivo in yeast. These results provide evidence for the response of *PeNAC* members under cold stress. In physical and chemical property terms, the isoelectric point of the PeNAC protein ranges from 3.84 to 10.44, with an average of 6.38, which is consistent with *BjuNAC* [47]. All *PeNAC* family members are subcellularly localized in the nucleus by prediction. This was consistent with the localization of most transcription factors.

In general, the expression level of a gene determines its function [48]. Transcription factors usually play a key role in regulating the expression of tissue-specific genes [49]. In this study, we have selected some *PeNAC* genes for qRT-PCR analysis of each tissue, and the results show that some genes exhibited higher expression in the roots, such as *PeNAC-4/13/20*. Similar results were also shown in other plants, such as orchardgrass (*Dactylis glomerata* L.) [45], *Fagopyrum tataricum* [50], *Panicum miliaceum* [51] and *Triticum aestivum* [52]. In orchards, *DgNAC046/087/103* had the highest expression in the stems, which may play an important role in stem development. In this research, the expression of *PeNAC-6/19/17/23* is higher in flowers, which may be related to the development of floral organs. In addition, the overexpression of the tissue’s specifically expressed NAC gene, poplar *NAC15*, was able to promote wood formation [53]. In orchardgrass, *DgNAC* genes are extensively involved in tissue development [45]. The NAC domain transcription factor *PdWND3A* affects lignin biosynthesis [54].

The relationship between NAC family members and plant abiotic stress has also been reported in many species. In *B. juncea* var. *tumida*, some NAC genes such as *BjuNAC112*, *BjuNAC178*, *BjuNAC184* and *BjuNAC240* can respond to high-temperature stress [55]. In addition, in tobacco, the overexpression of *LpNAC13* in *Lilium pumilum* reduced the tolerance to drought stress, but positively regulated the response to salt stress [47]. In *Cerasus humilis*, *ChNAC1* positively regulates the expression of abscisic acid (ABA)-responsive genes, and its overexpressed plants exhibited higher drought tolerance [25,56]. Arabidopsis AtNTL6 expressed the highest expression at low temperature for 18 h [57]. *OsNAC6/SNAC2* overexpression could improve the drought, salt and low-temperature tolerance of rice [23,24]. SlNAC1 from *Suaeda liaotungensis* and *VvNAC1* from *Vitis vinifera* L. can positively regulate the cold resistance of transgenic plants [58,59]. In wheat (*Triticum aestivum* L.), *TaNAC2*/47/67 can improve cold tolerance in transgenic plants [60,61,62]. *GmNAC20* transgenic rice has stronger salt tolerance and cold tolerance by regulating the expression of abiotic stress-response genes [63]. 

In this research, one member of the *PeNAC*s, *PeNAC-19*, was induced in the different abiotic stresses. *PeNAC-19* was induced by four various abiotic stresses. It may be a candidate gene for stress-resistant breeding. At present, low temperatures have seriously endangered the development of passion fruit cultivation. Therefore, we first focus on the relationship between *PeNAC-19* and cold stress. The results show that *PeNAC-19* responded to cold stress significantly in tobacco and Arabidopsis, and could improve the low-temperature tolerance of yeast. This study lays a foundation for the functional study of *PeNAC* gene family members under fruit ripening and abiotic stress.

## 5. Conclusions

In this study, the NAC family members in passion fruit were identified and analyzed, and the transcriptome results at different fruit-ripening stages and abiotic stresses were verified by qRT-PCR. The expression of one of the NAC genes (*PeNAC-19*) was induced by cold stress. Further verification of this gene showed that it could enhance the ability of transgenic tobacco, Arabidopsis and yeast to resist cold stress. The results lay a good foundation for further studies on the ability of the passion fruit to resist to abiotic stresses.

## Figures and Tables

**Figure 1 plants-12-01393-f001:**
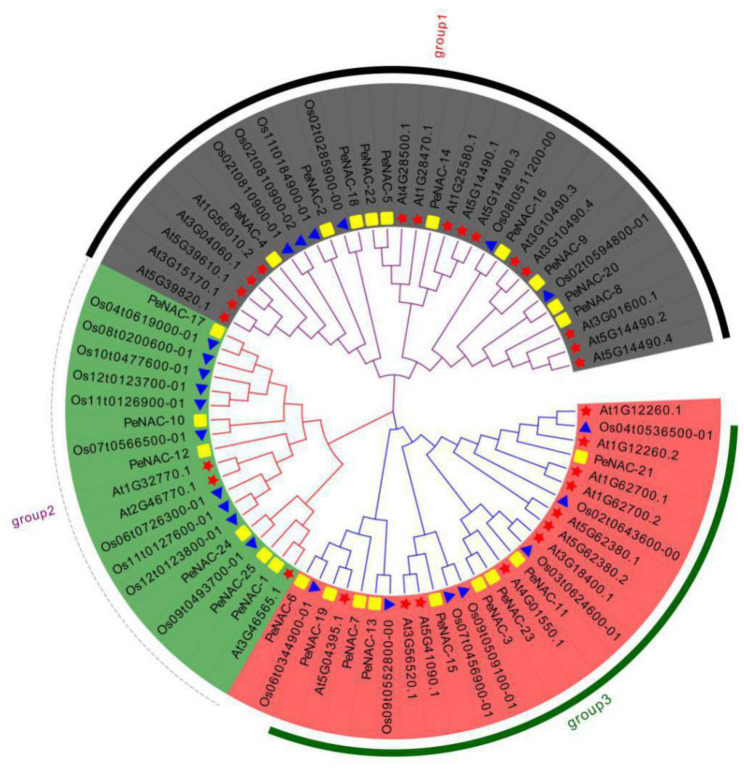
Phylogenetic evolutionary tree of NACs among the passion fruit, Arabidopsis and rice was constructed by ClustalX 2.0 and MEGA 7.0. The square represents the NACs in passion fruit; the five-pointed star represents the NACs in Arabidopsis; the triangle represents the NACs in rice.

**Figure 2 plants-12-01393-f002:**
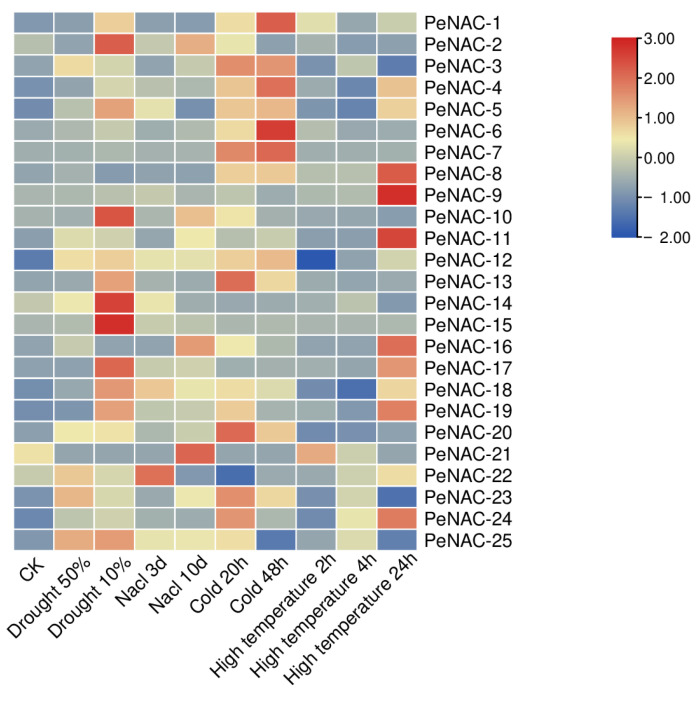
Expression profiles of *PeNACs* genes responding to drought (soil moisture of 50% and 10%), salt (300 mM NaCl for 3 d and 10 d), cold (0 °C for 20 h and 48 h) and high temperature (45 °C for 2 h, 4 h and 24 h). The plants of about 1 m and with 8–10 functional leaves were used for abiotic stress treatments. The details are shown in Appendix A. A low expression level is shown in blue and a high expression level is shown in red. CK means control check.

**Figure 3 plants-12-01393-f003:**
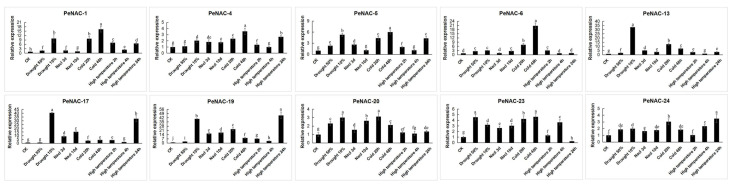
Expression analysis of 10 *PeNACs* under different abiotic stresses in the passion fruit. The experimental materials are identical to those used for RNA-seq under the abiotic stresses. The details are shown in Appendix A. Data are means ± SD of n = 3 biological replicates. Means denoted by the same letter are not significantly different at *p* < 0.05 as determined by Duncan’s multiple range test.

**Figure 4 plants-12-01393-f004:**
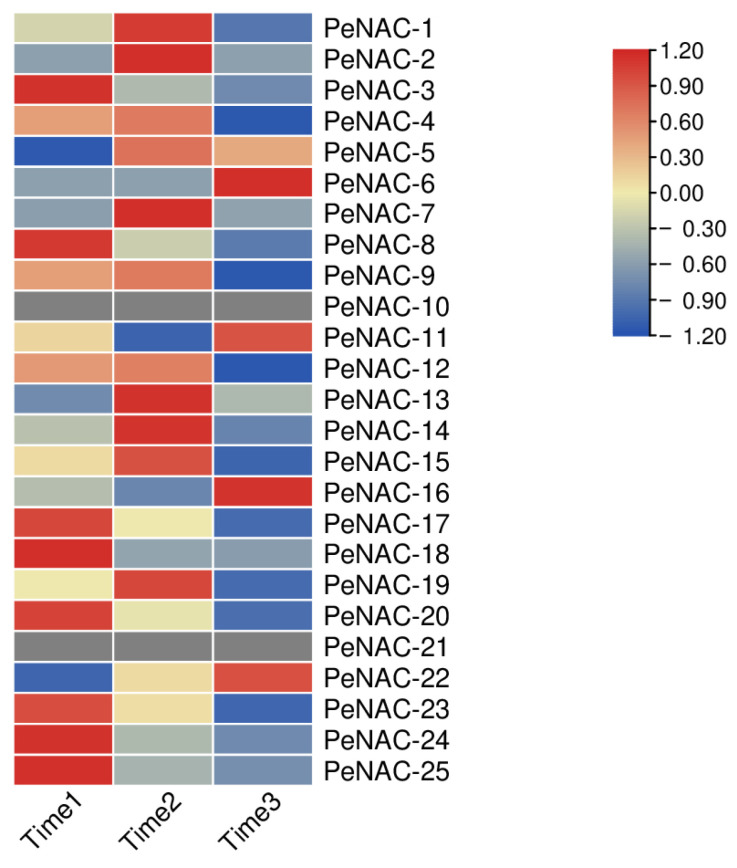
Expression profiles of *PeNACs* genes during three fruit-ripening periods (Time1: the 7th day before fruit harvest, Time2: the harvest period, Time3: the 7th day after fruit harvest). The details are shown in Appendix A. A low expression level is shown in blue and a high expression level is shown in red. The heat map was generated using TBtools.

**Figure 5 plants-12-01393-f005:**
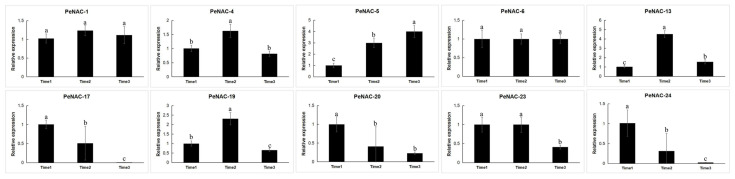
Expression analysis of 10 *PeNACs* during three fruit-ripening periods in the passion fruit. The experimental materials are identical to those used for RNA-seq during three fruit-ripening periods. The details are shown in Appendix A. Data are means ± SD of n = 3 biological replicates. Means denoted by the same letter are not significantly different at *p* < 0.05 as determined by Duncan’s multiple range test.

**Figure 6 plants-12-01393-f006:**
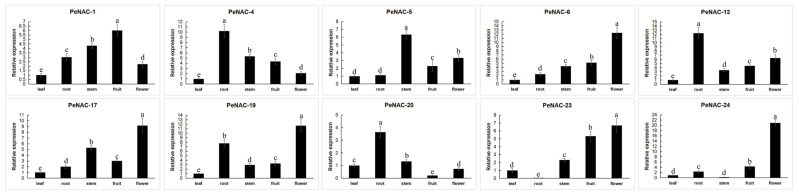
Expression analysis of 10 *PeNACs* in the leaf, root, stem, fruit and flower of the passion fruit. The leaf, root, stem, fruit, flower and fruits were obtained from the healthy passion fruit (*Passiflora edulis*). The details are shown in Appendix A. Data are means ± SD of n = 3 biological replicates. Means denoted by the same letter are not significantly different at *p* < 0.05 as determined by Duncan’s multiple range test.

**Figure 7 plants-12-01393-f007:**
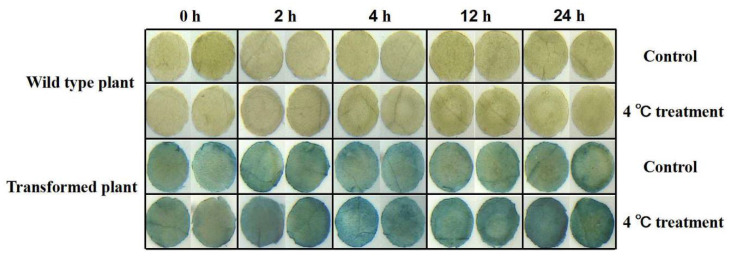
Gus staining was performed on transgenic tobacco under normal and cold treatment conditions. Tobacco was treated for 0, 2, 4, 12 and 24 h at low temperature (3 replicates per treatment) and then processed into discs (diameter 0.5 cm). The leaf discs that floated under normal growth temperature (25 °C) were used as control.

**Figure 8 plants-12-01393-f008:**
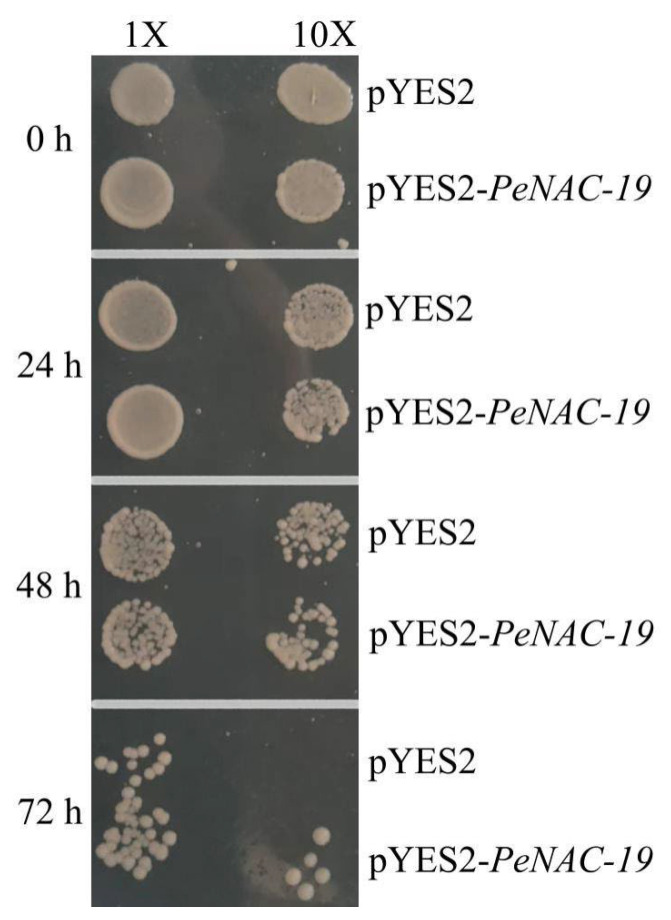
Growth status of the *Saccharomyces cerevisiae* INVSc1 strain expressing pYES2–*PeNAC-19* and pYES2 (control) under cold stress (−20 °C). 1× means the original bacterial fluid, 10× is the fluid diluted 10 times.

**Figure 9 plants-12-01393-f009:**
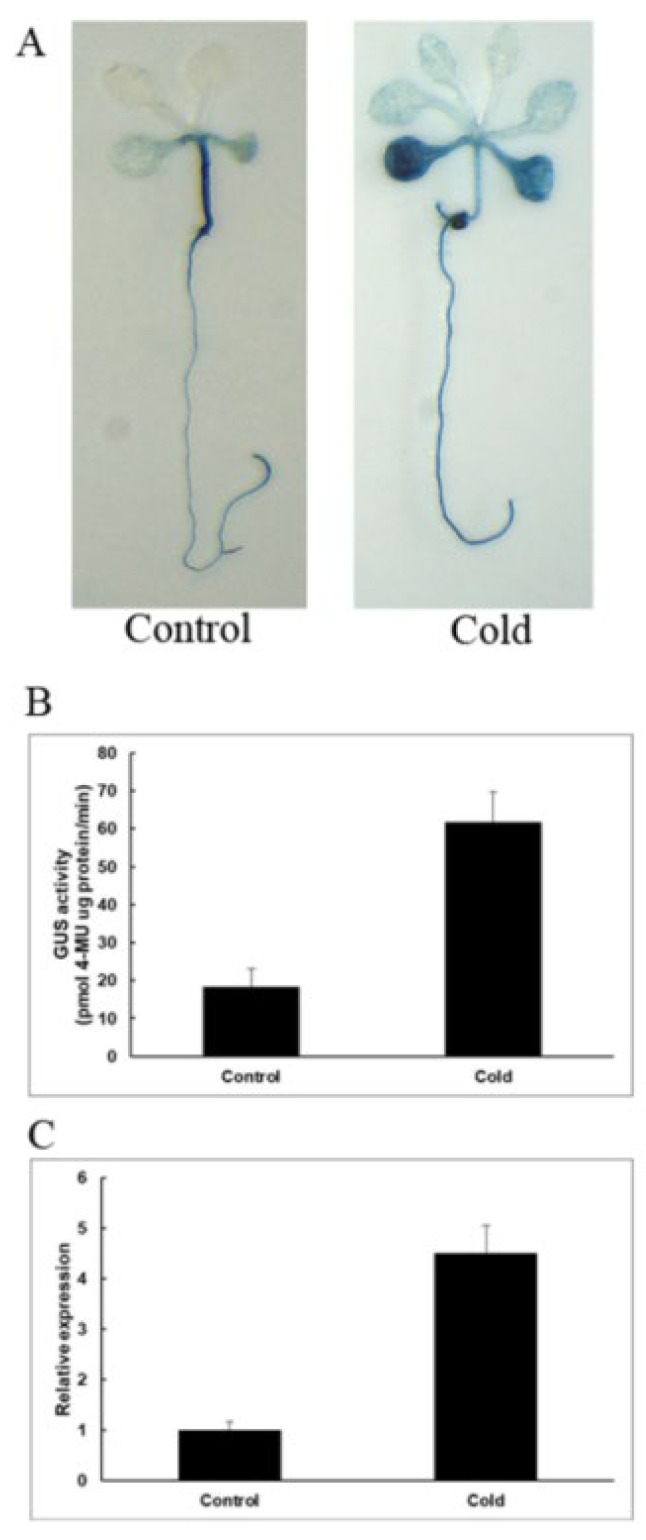
Induction and expression pattern of *PeNAC-19* under low temperature (4 °C for 36 h). (**A**) GUS staining of transgenic Arabidopsis. (**B**) GUS enzyme activity analysis of transgenic Arabidopsis thaliana. (**C**) Expression of *PeNAC-19* under the cold stress. Means denoted by the same letter are not significantly different at *p* < 0.05 as determined by Duncan’s multiple range test.

**Table 1 plants-12-01393-t001:** Basic information of NAC genes identified in passion fruit. MW: molecular weight. PI: isoelectric point.

Gene	Gene ID	CDS Length (bp)	Protein Length (aa)	MW (Da)	PI	Subcellular Localization
*PeNAC-1*	P_edulia010001655.g	1377	458	50.19	4.78	Nucleus
*PeNAC-2*	P_edulia010001845.g	786	261	29.7	8.76	Nucleus
*PeNAC-3*	P_edulia010003708.g	1524	507	56.98	6.28	Nucleus
*PeNAC-4*	P_edulia010004544.g	1938	645	72.96	4.61	Nucleus
*PeNAC-5*	P_edulia020006444.g	891	296	33.61	7.99	Nucleus
*PeNAC-6*	P_edulia030008739.g	927	308	35.6	8.44	Nucleus
*PeNAC-7*	P_edulia030009215.g	639	212	24.27	10.44	Nucleus
*PeNAC-8*	P_edulia030009267.g	1182	393	43.74	5.1	Nucleus
*PeNAC-9*	P_edulia030009488.g	975	324	35.12	3.84	Nucleus
*PeNAC-10*	P_edulia040010645.g	939	312	36.11	8.1	Nucleus
*PeNAC-11*	P_edulia050011226.g	468	155	17.98	4.41	Nucleus
*PeNAC-12*	P_edulia060013132.g	6126	2042	229.5	7.11	Nucleus
*PeNAC-13*	P_edulia060013466.g	594	197	22.43	10.39	Nucleus
*PeNAC-14*	P_edulia060013670.g	891	296	33.2	9	Nucleus
*PeNAC-15*	P_edulia060013771.g	576	191	22.03	4.92	Nucleus
*PeNAC-16*	P_edulia060014061.g	579	192	22.67	7.83	Nucleus
*PeNAC-17*	P_edulia060014082.g	246	81	9.47	4.57	Nucleus
*PeNAC-18*	P_edulia060015325.g	801	266	30.06	6.71	Nucleus
*PeNAC-19*	P_edulia060015528.g	840	279	30.93	4.39	Nucleus
*PeNAC-20*	P_edulia060015714.g	1284	427	48.17	4.89	Nucleus
*PeNAC-21*	P_edulia070017373.g	174	57	6.74	4.48	Nucleus
*PeNAC-22*	P_edulia080019184.g	927	308	34.59	7.69	Nucleus
*PeNAC-23*	P_eduliaContig140022926.g	1524	507	56.99	6.21	Nucleus
*PeNAC-24*	P_eduliaContig140022928.g	636	211	24.02	4.25	Nucleus
*PeNAC-25*	P_eduliaContig140022930.g	1551	516	57.88	4.49	Nucleus

## Data Availability

The raw sequence data have been deposited in the National Genomics Data Center (NGDC), and the link is https://ngdc.cncb.ac.cn/gwh/Genome/557/show accessed on 2 April 2021. All datasets are available from the corresponding author upon reasonable request.

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
