# Peer review of "Characterization of the NAC Transcription Factor in Passion Fruit (*Passiflora edulis*) and Functional Identification of *PeNAC-19* in Cold Stress"

_plants, 2023, doi:10.3390/plants12061393_

Round 1
Reviewer 1 Report
Title: Identification of the Passion fruit (Passiflora edulis) NAC family in fruit development and abiotic stress, and functional analysis of PeNAC-19 in cold stress
General comments: Please insert line numbers to the text document when submitting for the peer-review. I set up review lines for myself, and line 1 started with the title. Experimental objectives should be included. The materials and Methods section needs to cover the steps in the experimental procedure. Avoid using the word “it” and replace “it” with a specific corresponding noun as much as possible. Adding information explaining how the analysis is performed in figure titles is helpful; however, that information should be in the materials and methods section.
Formatting: Please check the formatting. There are multiple strikethrough lines, space (line 17, line 123, line 196, line 224, line 226, line 250, 272, 334), unnecessary italicized words (line 245), and repetition (line 339). Use the word to write integer numbers below eleven instead of numbers (lines 224 and 260: Not 3 subfamilies, but three subfamilies).
Introduction:
Lines 30: Consider changing the sentence for the grammar and correct word choice according to this context. “act a vital roles”
Lines 68: “It” plays an important role in the softening process [32]. What plays an important role? Please use a specific noun.
Lines 76-77: A grammar check is needed.
Lines 84-94: This paragraph should use to state the objectives of this study, not to summarize the study. Please reword these sentences to elaborate on how current work is similar/different from previous research and explain the objectives of the experiment. Also, explain the differences you observed between the two genomes. Please refrain from explaining current experiment results in the introduction.
Objectives:
The objectives of this study should be mentioned in the manuscript.
Materials and Methods:
Line 99: Please indicate the place where the genome is located. This link doesn’t guide you to the correct location.
Lines 97, 168, 223, 320, 322, 324, 328-330: Italicize the scientific name.
Line 105: Please include a correct link. The current link doesn’t direct to the database.
Lines 122-131: Different varieties have different genetic backgrounds, and abiotic stress mostly varies with genetic background. Therefore, explain the experimental design in detail, including the species and varieties you used for each abiotic stress treatment. Also, the number of replicates per treatment, the number of repetitions of each experiment, how plants were assigned to each treatment, and whether there is a transition period after treatment if you used the same plants for multiple treatments should be included the text.
Lines 133-137: Include how you proceed with the experiment during fruit development in the Materials and Methods.
Line 156 and Figure 7: explain WT in the figure title or use “wild type.”
Line 181: Describe plants with a specific height, number of leaves/nodes, etc..
Line 156, 337: Check for spelling
Results:
Lines 251-258 and 259-263: Includes information that goes to results, and Materials and Methods. Please separate them accordingly.
Line 317-318: Incomplete sentence.
Line 54, 320: Write genus and species names when you mention the scientific name for the first time, and then you can use the abbreviation for the genus.
Table 1: Explain abbreviations on the footnote.
Figure 2: What is CK? Explain all abbreviations before the first use.
Figure 3, 5, 6, 9B: Large letters and numbers are needed for easy reading.
Figure 8: What are 1X and 10X? Please explain dilution factors in the figure title. Also, state the (cold) temperature.
Conclusion: There is no conclusion in the manuscript.
Author Response
Thank you very much for your review of the manuscript, we have made corresponding amendments and additions, please check.If there are any more mistakes, please help us to point them out.
General comments: Please insert line numbers to the text document when submitting for the peer-review. I set up review lines for myself, and line 1 started with the title. Experimental objectives should be included. The materials and Methods section needs to cover the steps in the experimental procedure. Avoid using the word “it” and replace “it” with a specific corresponding noun as much as possible. Adding information explaining how the analysis is performed in figure titles is helpful; however, that information should be in the materials and methods section.
Formatting: Please check the formatting. There are multiple strikethrough lines, space (line 17, line 123, line 196, line 224, line 226, line 250, 272, 334), unnecessary italicized words (line 245), and repetition (line 339). Use the word to write integer numbers below eleven instead of numbers (lines 224 and 260: Not 3 subfamilies, but three subfamilies).
Thank you for your suggestion. The line nembers have been added in the manuscrtipt. We have reviewed all the manuscript and the formatting has been changed. If there are any errors that have not been corrected, please help us point them out.
Introduction:
Lines 30: Consider changing the sentence for the grammar and correct word choice according to this context. “act a vital roles”
Thank you for your suggestion. The relevant content has been changed in line 33.
Lines 68: “It” plays an important role in the softening process [32]. What plays an important role? Please use a specific noun.
Thank you for your suggestion. The relevant content has been changed in line 74.
Lines 76-77: A grammar check is needed.
Thank you for your suggestion. The relevant content has been changed in line 82.
Lines 84-94: This paragraph should use to state the objectives of this study, not to summarize the study. Please reword these sentences to elaborate on how current work is similar/different from previous research and explain the objectives of the experiment. Also, explain the differences you observed between the two genomes. Please refrain from explaining current experiment results in the introduction.
Thank you for your suggestion. he relevant content has been added in this part.
Objectives:
The objectives of this study should be mentioned in the manuscript.
Thank you for your suggestion. The objectives of this study should be mentioned in the abstract.
Materials and Methods:
Line 99: Please indicate the place where the genome is located. This link doesn’t guide you to the correct location.
Thank you for your suggestion. The link has been changed to the latest one and added in line 106.
Lines 97, 168, 223, 320, 322, 324, 328-330: Italicize the scientific name.
Thank you for your suggestion. We have italicized all the scientific name in the manuscript.If there are any omissions, please point them out.
Line 105: Please include a correct link. The current link doesn’t direct to the database.
Thank you for your suggestion. Because the line number has been added, some information does not correspond. Does this link refer to (http://plants.ensembl.org/index.html)?If so, the link can be replaced with http://plants.ensembl.org/Arabidopsis_thaliana/Info/Index and http://plants.ensembl.org/Oryza_sativa/Info/Index.
Lines 122-131: Different varieties have different genetic backgrounds, and abiotic stress mostly varies with genetic background. Therefore, explain the experimental design in detail, including the species and varieties you used for each abiotic stress treatment. Also, the number of replicates per treatment, the number of repetitions of each experiment, how plants were assigned to each treatment, and whether there is a transition period after treatment if you used the same plants for multiple treatments should be included the text.
Thank you for your suggestion. The relevant content has been changed in section 2.3.
Lines 133-137: Include how you proceed with the experiment during fruit development in the Materials and Methods.
Thank you for your suggestion. This section is described in the results in line 279-281. Also it has been added in line 141-145.
Line 156 and Figure 7: explain WT in the figure title or use “wild type.”
Thank you for your suggestion. The relevant content has been changed in line 171 and the legend of figure 7.
Line 181: Describe plants with a specific height, number of leaves/nodes, etc..
Thank you for your suggestion. The relevant content has been changed in line 192.
Line 156, 337: Check for spelling
Thank you for pointing out the mistake. We have modified it, if there are other errors, please help to point out.
Results:
Lines 251-258 and 259-263: Includes information that goes to results, and Materials and Methods. Please separate them accordingly.
Thank you for your suggestion. The relevant content has been changed in part 3.3.
Line 317-318: Incomplete sentence.
Thank you for pointing out the mistake. The relevant content has been changed in line 317-318.
Line 54, 320: Write genus and species names when you mention the scientific name for the first time, and then you can use the abbreviation for the genus.
Thank you for your suggestion. The relevant content has been changed in line 57,59,323.
Table 1: Explain abbreviations on the footnote.
Thank you for your suggestion. The relevant content has been added.
Figure 2: What is CK? Explain all abbreviations before the first use.
Thank you for your suggestion. The relevant content has been changed in the legend of figure 2.
Figure 3, 5, 6, 9B: Large letters and numbers are needed for easy reading.
Thank you for your suggestion. The font and numbers are not clear because the pictures have been shrunk. We have uploaded a clear pictures of the original in the system.Please check.
Figure 8: What are 1X and 10X? Please explain dilution factors in the figure title. Also, state the (cold) temperature.
Thank you for your suggestion. The relevant content has been added in the legend of figure 8.
Conclusion: There is no conclusion in the manuscript.
Thank you for your suggestion. The conclusion has been added in the manuscript.
Reviewer 2 Report
In this manuscript (plants-2264327) entitled "Identification of the Passion fruit (Passiflora edulis) NAC family in fruit development and abiotic stress, and functional analysis of PeNAC-19 in cold stress" submitted to Plants, Yi Xu and colleagues characterized NAC family members in passion fruit and found that PeNAC-19 was induced by various abiotic stresses. In addition, authors revealed that PeNAC-19 could respond to cold stress significantly in tobacco and Arabidopsis, and could improve the low temperature tolerance of yeast. The data are convincing and the writing is clear and straightforward. However, some issues need to be addressed for improving the quality of this manuscript.
1, The title "Identification of the Passion fruit (Passiflora edulis) NAC family in fruit development and abiotic stress, and functional analysis of PeNAC-19 in cold stress" is convoluted. Please rephrase.
2, For Table 1, genes listed here should be submitted to the NCBI database.
3, For Figure 2 and Figure 4, plant materials and treatments should be described in the legend.
4, For Figure 3, Figure 5 and Figure 6, important informations about these qRT-PCR analyses, including plant materials, treatments and significant difference analysis, should be described in their legends
5, For Figure 7, the GUS color difference between controls and cold treatments is very small. Please quantify GUS values in the revision.
6, Please double-check the reference list. For instance, both journal abbreviations and full names appeared.
Author Response
Thank you very much for your review of the manuscript, we have made corresponding amendments and additions, please check.If there are any more mistakes, please help us to point them out.
1, The title "Identification of the Passion fruit (Passiflora edulis) NAC Transcription Factor in fruit development and abiotic stress, and functional analysis of PeNAC-19 in cold stress" is convoluted. Please rephrase.
Thank you for your suggestion. The title has been changed into “Characterization of the NAC Transcription Factor in Passion Fruit (Passiflora edulis) and functional identification of PeNAC-19 in cold stress”. If you have any better suggestions, please let us know.
2, For Table 1, genes listed here should be submitted to the NCBI database.
Thank you for your suggestion. These genes will be submitted to the NCBI.
3, For Figure 2 and Figure 4, plant materials and treatments should be described in the legend.
Thank you for your suggestion. The relevant content has been added in the legend of Figure 2 and Figure 4.
4, For Figure 3, Figure 5 and Figure 6, important informations about these qRT-PCR analyses, including plant materials, treatments and significant difference analysis, should be described in their legends
Thank you for your suggestion. The relevant content has been added in the legend of Figure 3, Figure 5 and Figure 6 .
5, For Figure 7, the GUS color difference between controls and cold treatments is very small. Please quantify GUS values in the revision.
Thank you for your suggestion. For the GUS staining, as shown in Figure 9A, under the normal growth condition, the staining of transgenic seedlings was mainly concentrated in the stem. Under the low temperature stress, the expression of gus gene was mainly in the leaves and roots. The whole plant has been used to detect GUS enzyme activity. And the GUS activities is 3.4-folds higher under the cold stress than the control.
6, Please double-check the reference list. For instance, both journal abbreviations and full names appeared.
Thank you for pointing out the mistake. All the journal names have been changed to full names.
Round 2
Reviewer 2 Report
Authors have addressed my concerns in the revised manuscript. It looks much improved and sutiable for publishing on the premier journal Plants.